# Self-Supervised Surround-View Depth Estimation with Volumetric Feature Fusion

**Jung-Hee Kim** *
42dot Inc.
junghee.kim@42dot.ai

**Junhwa Hur** [*,†]
Google Research
junhwahur@google.com

**Tien Phuoc Nguyen** [†]
Hyundai Motor Group Innovation Center
tien.nguyen@hmgics.com

**Seong-Gyun Jeong**
42dot Inc.
seonggyun.jeong@42dot.ai

## Abstract

We present a self-supervised depth estimation approach using a unified volumetric feature fusion for surround-view images. Given a set of surround-view images, our method constructs a volumetric feature map by extracting image feature maps from surround-view images and fuse the feature maps into a shared, unified 3D voxel space. The volumetric feature map then can be used for estimating a depth map at each surround view by projecting it into an image coordinate. A volumetric feature contains 3D information at its local voxel coordinate; thus our method can also synthesize a depth map at arbitrary rotated viewpoints by projecting the volumetric feature map into the target viewpoints. Furthermore, assuming static camera extrinsics in the multi-camera system, we propose to estimate a canonical camera motion from the volumetric feature map. Our method leverages 3D spatio-temporal context to learn metric-scale depth and the canonical camera motion in a self-supervised manner. Our method outperforms the prior arts on DDAD and nuScenes datasets, especially estimating more accurate metric-scale depth and consistent depth between neighboring views.

## 1 Introduction

Depth perception of surrounding environment is one of the key components for 3D vision applications such as autonomous driving, robotics, or augmented reality. Specifically for autonomous driving, 3D perception benefits numerous downstream tasks including 3D objects detection [7, 29, 46], 6D pose estimation [6, 21], object tracking [18, 41], etc.

While human drivers can proactively move their heads and eyes to observe their surrounding environment, an autonomous vehicle equips multiple cameras with fixed viewpoints and monitors its surroundings. The multi-camera system hence exhibits limitations subjective to the fixed camera setup; the camera system might share a small portion between the adjacent viewpoints and rely on heterogeneous camera intrinsics.

As a prior work, Full Surround Monodepth (FSM) [15] extended a self-supervised monocular depth estimation method [13] into a multi-camera setup of autonomous vehicles. Their method exploits small image overlaps between spatio-temporally neighboring cameras as a supervision signal to learn metric-scale depth. Yet, FSM [15] individually estimates depth and camera motion for each view with a shared CNN, and it does not utilize any additional image cues from neighboring views at test time.

---

*denotes equal contribution. [†] This work has been done at 42dot Inc.

36th Conference on Neural Information Processing Systems (NeurIPS 2022).

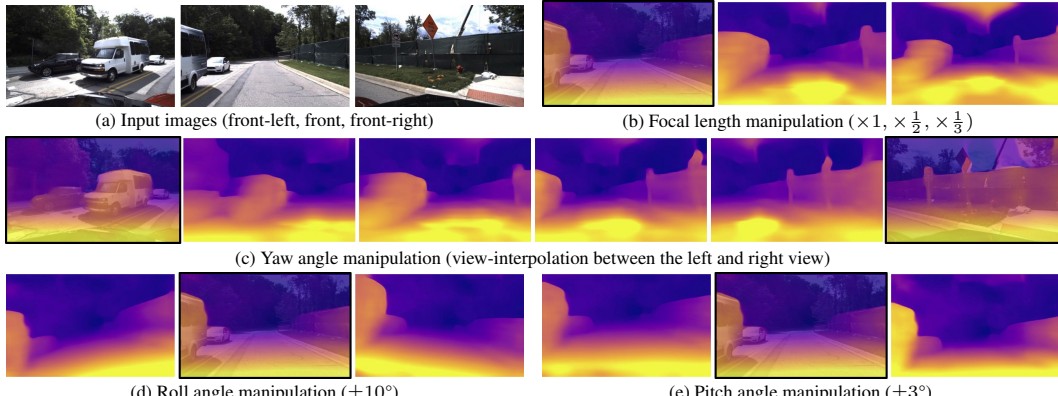

(a) Input images (front-left, front, front-right)

(b) Focal length manipulation ($\times 1, \times \frac{1}{2}, \times \frac{1}{3}$)

(c) Yaw angle manipulation (view-interpolation between the left and right view)

(d) Roll angle manipulation ($\pm 10°$)

(e) Pitch angle manipulation ($\pm 3°$)

Figure 1: **Visualization of synthesized depths**: *(a)* Given a set of images (e. g. front-left, front, and front-right), we can synthesize depth map at imaginary views with different *(b)* focal lengths (zoom-out effect), *(c)* yaw angles, *(d)* roll angles, and *(e)* pitch angles. Depth maps at known camera views are overlayed with input images.

To date, architectural design choices in the spatio-temporal domain are relatively less investigated, creating room for improvements.

We introduce a self-supervised approach to surround-view depth estimation based on a unified volumetric feature representation. Given multiple images covering the surrounding view, our method extracts an image feature from each image, back-projects the features into 3D space (*i. e.*, voxel coordinate), and fuses them into a unified volumetric feature map. Each volumetric feature contains 3D information at its located voxel coordinate. Some volumetric features are shared between neighboring views due to their image overlaps; we present a tailored multilayer perceptron (MLP) to process the volumetric features with consideration of the superposition. Given the volumetric features, we project the features back to the image coordinate with known camera information and pass through the depth decoder to obtain a depth map at each view.

Because the volumetric feature contains 3D information at its local voxel coordinate, our method can also synthesize a depth map at arbitrary rotated viewpoints by projecting the features into an image coordinate with a target focal length and target extrinsics. This can overcome the limitation of fixed multi-camera setup. Fig. 1 demonstrates the results of depth map synthesis at novel views. Our method can seamlessly synthesize depth maps at arbitrary viewpoints from the volumetric features with different focal lengths and camera angles.

Furthermore, unlike previous work [15], we introduces a canonical camera motion estimation, assuming static extrinsics between cameras. This geometry constraint leads to a stabilized ego-motion learning and enhanced metric-scale depth estimation. On top of that, we adjust the intensity distribution for the neighboring viewpoints, which reduces irregular photometric reconstruction errors. We present a self-supervised learning framework that learns to estimate scale-aware depth and camera motion from unlabeled surround-view images, using 3D spatio-temporal reconstruction.

We summarize our contributions as follows: (i) we introduce a novel volumetric feature representation that effectively learns to estimate the surround-view depth and canonical camera motion. To our knowledge, our method is the first approach that targets multi-camera feature fusion with self-supervised depth estimation. (ii) We demonstrate synthesizing depth maps at arbitrary views with the unified volumetric feature representation. (iii) On popular multi-camera datasets [2, 14], our method demonstrates consistent improvements compared to the state-of-the-art method [15].

## 2    Related Work

**Monocular depth estimation.**    Supervised-learning-based monocular depth approaches [8, 10, 20, 22, 36] require a large amount of annotated data for supervision, and as a result, it limits the scalability of their methods due to the expense of acquiring such annotated data on diverse scenes. To address the limitation, self-supervised methods propose proxy learning tasks that use unlabeled temporally-consecutive images [3, 14, 23, 24, 37, 47, 49, 51, 52] or stereoscopic pairs [11–13, 25, 34, 35] during

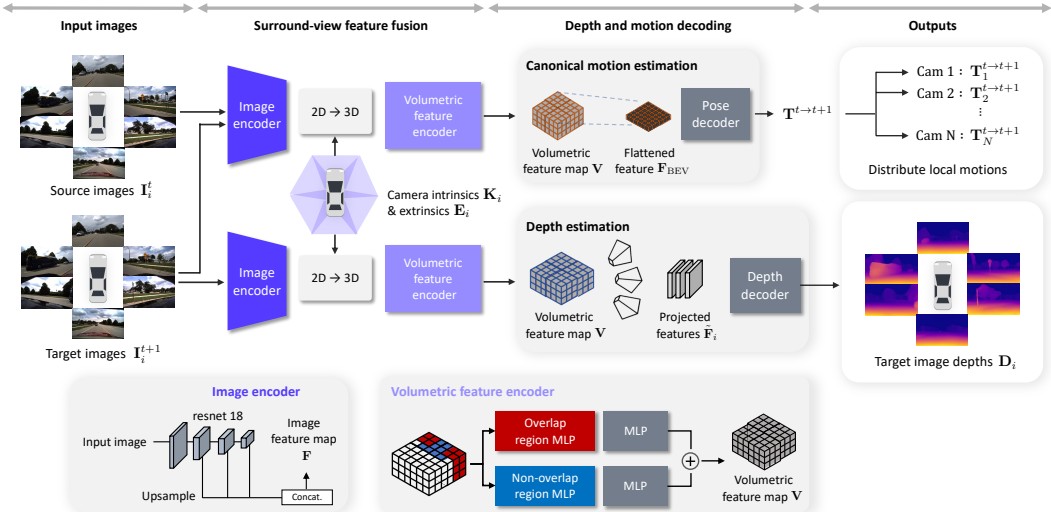

Figure 2: **Architecture overview**: Our method first extracts image feature maps from surround-view images and fuses them into into a volumetric feature map by considering overlapped features between neighboring views. The depth estimation module projects the volumetric feature into each view and estimates per-view depth map. The canonical motion estimation module flattens the volumetric feature along the z-axis and outputs the canonical motion. The output canonical motion can be distributed to each view as local motions.

training time. For supervision signals, the approaches minimize image reconstruction errors between reference images and synthesized images that are generated from output depths and temporally-(or spatially-)neighboring images. Yet, the methods output depth maps only up to scale or at a fixed viewpoint. In this work, we demonstrate a surround-view depth method that outputs depth maps at arbitrary viewpoints as well as in a metric scale.

**Omnidirectional depth estimation.** Previous work has demonstrated estimating depth on wide field-of-view (FOV) images [48] or 360° images [45, 44, 53] that covers surrounding view, and efforts on finding a suitable projective geometry to process those input images have been continued. Initial approaches [45, 53] directly used 360° images that are represented under equirectangular projection, but there exists severe visual distortion on the top and bottom parts of the images. To avoid such distortion on the 360° image, a cubemap representation [4, 43, 44] has been presented, which rectifies a input 360° image into a cube map coordinate and uses it for the input instead. Won et al. [48] proposed to directly perform a stereo matching of multiple fisheye images in a spherical coordinate. Neural Ray Surfaces [42] introduced a generic framework that jointly learns to estimate depth and motion without prior knowledge of specific camera models.

**Volumetric feature representation.** The basic idea of fusing multi-view features into a shared voxel space has been presented in other tasks, such as multi-view stereo [19, 26, 30, 32, 50] or 3D semantic segmentation [5, 26]. The approaches mainly focus on solving a matching task for object-level or static indoor scene reconstruction using intractable 3D convolution. Based on the similar principle, our method presents a volumetric feature representation for surround-view depth and canonical motion estimation. Comparing to 3D convolution, we demonstrate that light-weighted MLP layers are more efficient for fusing features from surround-view images with small overlaps.

Some approaches [17, 28, 33, 38] demonstrate aggregating 2D features from surround-view images onto a 2D Bird's-eye-view (BEV) space for semantic segmentation or object detection. However, the BEV-based representation does not precisely preserve 3D information due to the abstraction of the height information. In contrast, our approach presents a voxel-based representation that is more suitable for depth and canonical pose estimation.

# 3 Surround-View Depth Estimation via Volumetric Feature Fusion

Given two temporally consecutive surround-view images, $\mathbf{I}_i^t$ and $\mathbf{I}_i^{t+1}$, captured by multiple cameras $C_i$ ($i \in \{1, 2, ..., 6\}$ in our setup) that cover a surrounding view of a vehicle, our method estimates metric-scale depth $\mathbf{D}_i^t$ for each image $\mathbf{I}_i^t$ from each camera $C_i$ (but also at an arbitrary rotated viewpoint) and a canonical camera motion of the vehicle $\mathbf{T}^{t \to t+1}$ as an auxiliary task. Our methods assumes known camera intrinsics $\mathbf{K}_i$ and extrinsics $\mathbf{E}_i$, but each camera can have different intrinsics.

## 3.1 Architecture overview

We propose a novel fusion module for surround-view depth estimation. Fig. 2 shows an overview of our approach, consisting of the surround-view volumetric feature fusion (Sec. 3.2), depth estimation (Sec. 3.3) and canonical motion estimation (Sec. 3.4) module.

The surround-view feature fusion module constructs a single, unified volumetric feature map $\mathbf{V}$ by aggregating multi-scale image feature maps $\mathbf{F}$ that are extracted from each image in the surround-view camera setup. From the volumetric feature map $\mathbf{V}$, the depth estimation module retrieves a per-frame feature that represents a specific viewpoint and passes it through a depth decoder to produce a depth map for the corresponding view. Because a volumetric feature at each voxel contains its local 3D information, our method can also synthesize a depth map at an arbitrary view by projecting the volumetric feature map into the target view. The canonical motion estimation module flattens the volumetric feature map along the z-axis and predicts a global motion referring to the canonical camera coordinate. In contrast to FSM [15] that independently estimates each camera motion, our design globally reasons a camera motion, which not only reduces the number of unknown factors but also makes the problem simpler.

## 3.2 Surround-view volumetric feature fusion

**Image feature encoding.**    We first extract image features from surround-view images. Given a set of surround-view images, we pass each image $\mathbf{I}_i$ through a shared 2D image encoder to obtain a image feature map $\mathbf{F}_i$ for $i^{\text{th}}$ view. We use ResNet [16] as the encoder which takes an image $\mathbf{I}_i$ (with a resolution of $H \times W$) and outputs a multi-scale feature pyramid. We take the last three feature maps from the feature pyramid, reshape them up to a resolution of $\frac{H}{8} \times \frac{W}{8}$, concatenate them, and apply $1 \times 1$ convolution for channel reduction. As a result, we obtain a single feature map $\mathbf{F}_i$ for each image, where the feature map conveys multi-scale and high-dimensional information of the image.

**Volumetric feature encoding.**    We aggregate the image feature maps $\mathbf{F}_i$ from surround-view images into a single, unified volumetric feature map $\mathbf{V}$ in a pre-defined voxel space, as illustrated in Fig. 3. For each voxel, we find a corresponding pixel $\mathbf{p}(w, h)$ by projecting the voxel $(x, y, z)$ into a pixel coordinate. Then, we bilinearly interpolate an image feature $\mathbf{F}(\mathbf{p})$ at the projected pixel $\mathbf{p}(w, h)$ to handle sub-pixel location and allocate the sampled image feature to the voxel. Because the sampled image feature $\mathbf{F}(\mathbf{p})$ contains high-level information along its pixel ray, we extract local 3D feature at the voxel $(x, y, z)$ by concatenating the feature with a depth value of the voxel (*i.e.*, depth positional encoding) and pass it through an MLP. In this way, voxels that get through by a single pixel ray have their individual 3D feature extracted by the MLP instead of having the same image feature.

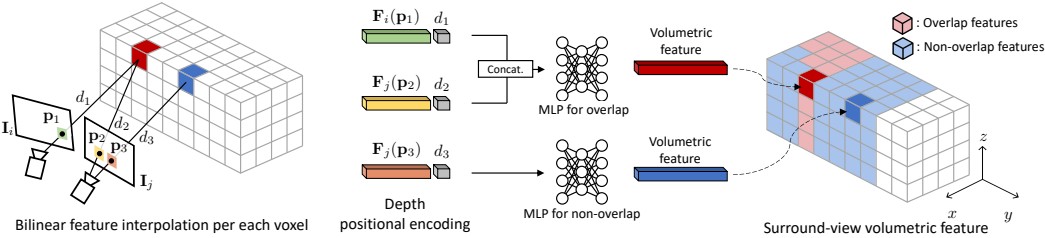

Figure 3: **Surround-view volumetric feature fusion**: For each voxel, we first bilinearly interpolate corresponding image features from each view. After the depth positional encoding, we pass the features through MLPs to encode volumetric feature. We use two different MLPs for overlap and non-overlap region.

Some voxels are associated with multiple image features due to spatial overlaps across the camera views. To merge the multiple features on overlap region, we use the other MLP that learns to fuse the multiple per-pixel image features into a per-voxel volumetric feature.

## 3.3 Depth estimation

From the unified volumetric feature map, we aim to estimate a depth map of each surround-view input image but also synthesize depth maps at any arbitrary rotated camera viewpoints.

**Transformation of volumetric feature as projected image feature.** For each target view $C_i$ that we want to estimate its depth map, we project the volumetric feature map $\mathbf{V}$ into the projected image feature map $\tilde{\mathbf{F}}_i$ with a resolution of $\frac{H}{8} \times \frac{W}{8}$, using camera extrinsics $\mathbf{E}_i$ of the target view. For each pixel $\mathbf{p}$ at the target image coordinate, we uniformly sample volumetric features along its pixel ray and concatenate the sampled features to obtain the projected image feature $\tilde{\mathbf{F}}_i(\mathbf{p})$.

**Depth decoder.** To predict a depth map from the projected image feature map $\tilde{\mathbf{F}}_i$, we use a light-weighted depth decoder $Dec_{\text{depth}}$ consisting of 4 convolutional layers: 3 convolutional layers for upsampling ($\frac{H}{8} \times \frac{W}{8} \to H \times W$) and one convolutional layer for the depth output.

**Depth map synthesis at a novel view.** We further extend our depth estimation module to produce a depth map for a novel viewpoint which is not covered by the original multi-camera system. Because a volumetric feature at each voxel encodes local 3D structure information in a certain range, our method can synthesize a depth map at a desired arbitrary viewpoint by projecting the volumetric feature into the viewpoint and passing it through the depth decoder. Fig. 1 visualizes synthesized depth maps at arbitrary rotated viewpoints. Our method can seamlessly synthesize depth maps with variations of yaw, roll, or pitch angles as well as arbitrary focal lengths.

## 3.4 Canonical motion estimation

The prior work, FSM [15], separately estimates a relative camera motion of each camera using a shared encoder-decoder network. However, this process can be redundant because a multi-camera system that is attached to a vehicle shares the same canonical motion. To improve, we introduce a canonical pose estimation module that takes a collapsed volumetric feature map $\mathbf{F}_{\text{BEV}}$ and predicts one representative canonical camera motion as illustrated in Fig. 2. Given the estimated canonical camera motion, we can direct compute the motion of each camera with their known camera extrinsics. Here, we use the same architecture design to obtain a volumetric feature map (cf. Sec. 3.2) but with different trainable network weights, and use two temporally consecutive images for the input.

**Volumetric feature reduction.** Given a surround-view volumetric feature map $\mathbf{V} \in \mathbb{R}^{\mathbb{X} \times \mathbb{Y} \times \mathbb{Z} \times \mathbb{C}}$, we flatten the feature map onto a Bird's-Eye-View (BEV) shape (see Fig. 2). We collapse the $\mathbf{Z}$ dimension into the channel dimension $\mathbf{C}$ (*i.e.*, reshaping it into a 3D tensor $\mathbf{V}' \in \mathbb{R}^{\mathbb{X} \times \mathbb{Y} \times (\mathbb{Z} \cdot \mathbb{C})}$) and apply 2D convolutions to reduce channel of the feature, resulting in a 3D tensor of $\mathbf{F}_{\text{BEV}} \in \mathbb{R}^{\mathbb{X} \times \mathbb{Y} \times \mathbb{C}'}$. This collapsed volumetric feature map $\mathbf{F}_{\text{BEV}}$ contains information on canonical ego-motion that is aggregated from surrounding views. Then we use the standard pose decoder from the PoseNet [13] to estimate the canonical camera motion $\mathbf{T}^{t \to t+1}$.

**Computing local camera poses.** Assuming a static relationship between the cameras, we distribute the predicted canonical camera motion to each camera motion. From the given extrinsic $\mathbf{E}_i$ of each camera $C_i$ and canonical camera motion $\mathbf{T}^{t \to t+1}$, we compute each camera motion as:

$$\mathbf{T}_i^{t \to t+1} = \mathbf{E}_i^{-1} \mathbf{E}_1 \mathbf{T}^{t \to t+1} \mathbf{E}_1^{-1} \mathbf{E}_i, \tag{1}$$

where $\mathbf{E}_1$ is the extrinsic of the canonical camera motion. Note that any viewpoints could also be a canonical motion. In this work, we define the canonical motion as the motion of the front-view camera.

## 3.5 Self-supervised learning

Given the camera motion and depth map per each view, we apply our self-supervised loss given a temporal triplet of surround-view images. At test time, our method only requires images at the current time step $t$.

**Self-supervised loss.** Our self-supervised loss consists of the image reconstruction loss $\mathcal{L}_{\text{img}}$ and the depth synthesis loss $\mathcal{L}_{\text{depth}}$:

$$\mathcal{L} = \mathcal{L}_{\text{img}} + \mathcal{L}_{\text{depth}}. \tag{2}$$

The image reconstruction loss penalizes the reconstruction error between the reference image $\mathbf{I}^t$ and the synthesized images $\tilde{\mathbf{I}}^{t+1}$ from each temporal, spatial, and spatio-temporal context (cf. Eq. (4)),

$$\mathcal{L}_{\text{img}} = \mathcal{L}_{\text{t}} + \lambda_{\text{sp}}\mathcal{L}_{\text{sp}} + \lambda_{\text{sp\_t}}\mathcal{L}_{\text{sp\_t}} + \lambda_{\text{smooth}}\mathcal{L}_{\text{smooth}}. \tag{3}$$

Each term stands for the temporal $\mathcal{L}_{\text{t}}$, spatio $\mathcal{L}_{\text{sp}}$, spatio-temporal $\mathcal{L}_{\text{sp\_t}}$, and smoothness loss $\mathcal{L}_{\text{smooth}}$. To measure the reconstruction error, we use a weighted sum of intensity difference and structure similarity [12, 13, 47]: $(1-\alpha)\|\mathbf{I}^t - \tilde{\mathbf{I}}^{t+1}\|_1 + \alpha\frac{1-\text{SSIM}(\mathbf{I}^t,\tilde{\mathbf{I}}^{t+1})}{2}$, with $\alpha = 0.85$. The smoothness term $\mathcal{L}_{\text{smooth}}$ follows an edge-aware 1st-order smoothness: $\mathcal{L}_{\text{smooth}} = \frac{1}{N}\sum_{\mathbf{p}}\sum_{k\in x,y}\nabla_k\mathbf{D}\cdot e^{-\|\nabla_k\mathbf{I}\|_1}$.

One main difference to the baseline work [15] is that we do not use the pose consistency loss because our method outputs one representative camera motion for all views. This reduces the number of unknowns and thus encourages convergence and better accuracy.

**Spatio-temporal context.** The core idea of the self-supervised proxy task [15] is to exploit small overlaps between spatially or/and temporally neighboring images for matching to provide supervisionary signal and scale information for camera motion and depth. For a reference image $\mathbf{I}_i^t$ from camera $C_i$ at a time step $t$, it synthesizes reconstructed images $\tilde{\mathbf{I}}_i^t$ from *(i)* spatial neighboring images $\mathbf{I}_j^t$ ($j$: indices of neighboring cameras), *(ii)* temporally neighboring images $\mathbf{I}_i^{t'}$ ($t' \in \{t+1, t-1\}$), and *(iii)* spatio-temporally neighboring images $\mathbf{I}_j^{t'}$, given the predicted camera motion. A pixel-wise warping operation for the image reconstruction is defined as:

$$\mathbf{\Pi}_{ij}^{t\rightarrow t'} = \mathbf{K}_j\mathbf{X}_{ij}^{t\rightarrow t'}\mathbf{D}_i\mathbf{K}_i^{-1} \tag{4a}$$

with

$$\mathbf{X}_{ij}^{t\rightarrow t'} = \begin{cases} \mathbf{T}_i^{t\rightarrow t'}, t' \in \{t-1, t+1\} & \text{for temporal context,} \\ \mathbf{E}_j\mathbf{E}_i^{-1} & \text{for spatio context,} \\ \mathbf{E}_j\mathbf{E}_i^{-1}\mathbf{T}_i^{t\rightarrow t'}, t' \in \{t-1, t+1\} & \text{for spatio-temporal context,} \end{cases} \tag{4b}$$

with estimated depth $\mathbf{D}_i$ and camera intrinsics $\mathbf{K}_i$ and $\mathbf{K}_j$. Here, $\mathbf{X}_{ij}$ is the camera motion between the two cameras depending on their spatio-temporal context.

By minimizing the photometric difference between the reference image and reconstructed image, a self-supervised loss function guides the network to output metric-scale depths and camera motions that can satisfy the overall contexts.

**Depth synthesis loss.** We introduce a depth synthesis loss for the successful depth synthesis at a novel view:

$$\mathcal{L}_{\text{depth}} = \lambda_{\text{cons}}\mathcal{L}_{\text{cons}} + \lambda_{\text{depth\_smooth}}\mathcal{L}_{\text{depth\_smooth}} \tag{5a}$$

$$\mathcal{L}_{\text{cons}} = \frac{1}{N}\sum_{\mathbf{p}}(|\mathbf{D}_i - \tilde{\mathbf{D}}_i|)/(\mathbf{D}_i + \tilde{\mathbf{D}}_i) \tag{5b}$$

$$\mathcal{L}_{\text{depth\_smooth}} = \frac{1}{N}\sum_{\mathbf{p}}(\nabla_x\tilde{\mathbf{D}}_i + \nabla_y\tilde{\mathbf{D}}_i). \tag{5c}$$

The depth consistency loss $\mathcal{L}_{\text{cons}}$ penalizes the depth difference between the synthesized depth $\tilde{\mathbf{D}}_i$ at the novel view and depth $\mathbf{D}_i$ at each known camera view $i$, adopted from Bian et al. [1]. The smoothness loss $\mathcal{L}_{\text{depth\_smooth}}$ [27] encourages the 1st-order smoothness of the synthesized depth map $\tilde{\mathbf{D}}_i$ at the novel view so that it regularizes a set of volumetric features retrieved at the novel view to improve 3D-awareness at each voxel coordinate. For the synthesis, we augment a depth map at a novel view for each known camera view $i$ and apply the depth synthesis loss.

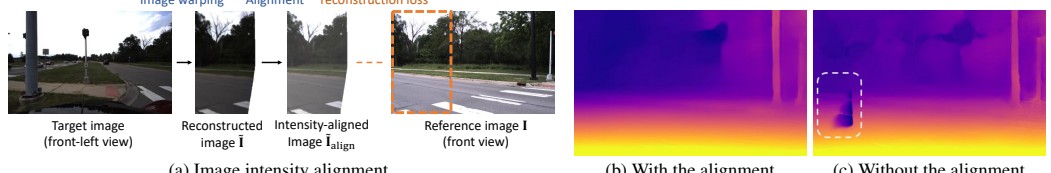

| | |
|---|---|
| Image warping   Alignment   reconstruction loss | |
| Target image          Reconstructed      Intensity-aligned    Reference image **I** | |
| (front-left view)      image **Ī**          Image **Ī**_align      (front view) | |
| (a) Image intensity alignment | (b) With the alignment    (c) Without the alignment |

Figure 4: **Intensity distribution alignment**: *(a)* After warping the target image, we align the mean and variance of the reconstructed image to match with reference image's. *(b)* When aligning the distribution, it resolves the artifacts on the depth, which appears *(c)* when training without the alignment.

# 4 Experiments

## 4.1 Implementation details

**Dataset and evaluation protocol.** We use the DDAD [14] and nuScenes [2] dataset for our experiments. Both datasets provide surround-view images from a total of 6 cameras mounted on a vehicle and LiDAR point clouds for the depth evaluation. We train our model on each train split and report the accuracy on the test split. During training, the input images are down-sampled to a resolution of $384 \times 640$ for the DDAD dataset, and that of $352 \times 640$ for the nuScenes dataset. We train our model on the DDAD dataset for 20 epochs and the nuScenes dataset for 5 epochs.

For evaluation, we follow the same protocol from FSM [15] that evaluates depth up to 200 $(m)$ for the DDAD dataset and 80 $(m)$ for the nuScenes dataset. We use conventional depth evaluation metrics proposed by Eigen et al. [8]. We provide details of the metrics in the supplementary material.

**Training.** We implemented our networks in PyTorch [31] and trained on four A100 GPUs. For the image encoders, we used ResNet-18 with weights pre-trained on ImageNet [39]. All experiments used the same training hyper-parameters (unless explicitly mentioned): Adam optimizer with $\beta_1 = 0.9$ and $\beta_2 = 0.999$; a mini-batch size of 2 per each GPU and a learning rate [40] with $1 \times 10^{-4}$, decaying at $\frac{3}{4}$ of the entire training schedule with a factor of 0.1; the previous $t - 1$ and subsequent $t + 1$ images are used as temporal context at the training time. Note that our depth estimation module does not require temporal contexts, and therefore temporal contexts are not included in the test time. For our volumetric feature, we used voxel resolution of (1m, 1m, 0.75m) with spatial dimensions of (100, 100, 20) for (x, y, z) axis respectively. We use color jittering as data augmentation. For the depth synthesis loss, we use the random rotation with a range between [-5°, -5°, -25°] and [5°, 5°, 25°] for the depth map synthesis at a novel view. In the self-supervised loss in Eq. (2), we use depth smoothness weight $\lambda_{\text{smooth}} = 1 \times 10^{-3}$, spatio loss weight $\lambda_{\text{sp}} = 0.03$, spatio-temporal weight $\lambda_{\text{sp\_t}} = 0.1$, depth consistency weight $\lambda_{\text{cons}} = 0.05$, and depth smoothness weight at novel views $\lambda_{\text{depth\_smooth}} = 0.03$.

The DDAD dataset [14] includes images with different focal lengths; in order to train the network to output a consistent depth scale regardless of different focal lengths, we use the focal length normalization [9] which normalizes the scale of output depth to a default focal length.

**Intensity distribution alignment.** We observe that different lighting conditions in neighboring images yields artifacts on the depth output (Fig. 4c) with the self-supervised loss in the previous work [15]. To address the issue, we align mean and variance on overlapped regions of the reference image **I** and synthesized image $\tilde{\mathbf{I}}$:

$$\tilde{\mathbf{I}}_{\text{align}} = (\tilde{\mathbf{I}} - \tilde{\mu}) \cdot \sigma/\tilde{\sigma} + \mu, \tag{6}$$

where $\{\mu, \sigma^2\}$ are the mean and variance of the reference image **I**, and $\{\tilde{\mu}, \tilde{\sigma}^2\}$ are that of the synthesized image $\tilde{\mathbf{I}}$. The aligned image $\tilde{\mathbf{I}}_{\text{align}}$ is then used for the loss calculation in Eq. (2). Fig. 4a briefly illustrates this process. Here, the reconstructed image can be from either a spatial, temporal, or spatio-temporal context.

## 4.2 Surround-view depth evaluation

We compare the accuracy of our method with state-of-the-art methods. Especially for our closest baseline FSM [15], we use our own reproduced version of FSM for the comparison because the

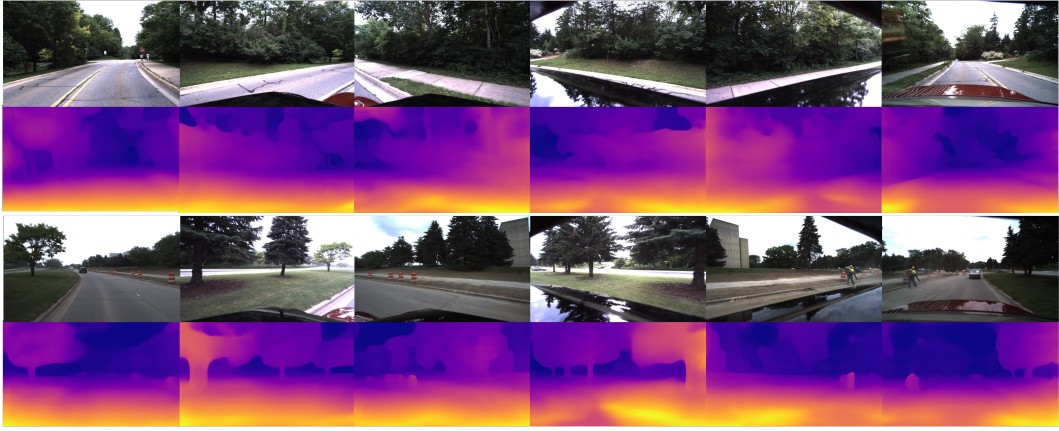

Figure 5: **Our results on DDAD dataset**: Our method successfully estimates depth at each view via volumetric feature fusion.

Table 1: **Evaluation on the DDAD dataset [14] and nuScenes dataset [2]**. We compare our method with the state of the arts. We report the averaged accuracy from all views using the metric from Eigen et al. [8]. Median-scale accuracy uses ground truth for correctly scaling the depth map during evaluation [8].

| Dataset | Scale | Method | Abs Rel | Sq Rel | RMSE | RMSE log | $\delta < 1.25$ | $\delta < 1.25^2$ | $\delta < 1.25^3$ |
|---|---|---|---|---|---|---|---|---|---|
| DDAD | Median | Monodepth2 [13] | 0.362 | 14.404 | 14.178 | 0.412 | 0.683 | 0.859 | 0.922 |
| | | PackNet-SfM [14] | 0.301 | 5.339 | 14.115 | 0.395 | 0.624 | 0.828 | 0.908 |
| | | Our reproduced FSM[15] | **0.219** | 4.161 | 13.163 | 0.327 | **0.703** | **0.880** | **0.940** |
| | | **Ours** | 0.221 | **3.549** | **13.031** | **0.323** | 0.681 | 0.874 | **0.940** |
| | Metric | Our reproduced FSM [15] | 0.228 | 4.409 | 13.433 | 0.342 | **0.687** | **0.870** | **0.932** |
| | | **Ours** | **0.218** | **3.660** | **13.327** | **0.339** | 0.674 | 0.862 | **0.932** |
| nuScenes | Median | Monodepth2 [13] | 0.287 | 3.349 | **7.184** | **0.345** | 0.641 | 0.845 | 0.925 |
| | | PackNet-SfM [14] | 0.309 | **2.891** | 7.994 | 0.390 | 0.547 | 0.796 | 0.899 |
| | | Our reproduced FSM [15] | 0.301 | 6.180 | 7.892 | 0.366 | **0.729** | 0.876 | 0.933 |
| | | **Ours** | **0.271** | 4.496 | 7.391 | 0.346 | 0.726 | **0.879** | **0.934** |
| | Metric | Our reproduced FSM [15] | 0.319 | 7.534 | 7.860 | 0.362 | **0.716** | 0.874 | 0.931 |
| | | **Ours** | **0.289** | **5.718** | **7.551** | **0.348** | 0.709 | **0.876** | **0.932** |

implementation is not provided. For the reproduction of FSM [13], we found that the intensity distribution alignment and focal length normalization, which we explained above, were critical for stable self-supervised training. Despite our efforts, there is still a gap between our reproduced version and the reported accuracy; for a fair comparison, we conduct experiments under the same training and evaluation settings and demonstrate accuracy gain over the reproduced baseline.

Table 1 demonstrates the evaluation on both DDAD [14] and nuScenes [2] datasets. We also report the accuracy using per-frame median scaling for the reference. On the DDAD dataset, our method shows competitive or better accuracy to the baseline method FSM [15] on 5 out of 7 metrics. Especially, our method substantially improves the accuracy on the *Sq Rel* metric by around 17%. On nuScenes dataset, our method demonstrates consistent improvement over the baseline, by outperforming the baseline on 6 out of 7 metrics, with over 24% improvement on the *Sq Rel* metric. Better accuracy is also observed when using the median scaling for evaluation. We further demonstrate point cloud reconstruction visualization in *Sec D.3* of supplementary material to provide clear evidence of the strength of our method.

### 4.3 Ablation study

**Surround-view volumetric feature fusion.** To validate our main contributions, we conduct an ablation study of our volumetric feature fusion and canonical motion estimation module over the FSM baseline [15]. We train the model on the DDAD dataset [14] while keeping the original experiment setup. As shown in Table 2, both of our contributions consistently improve the accuracy over the baseline. Our volumetric feature representation embeds effective 3D information and results in better depth accuracy; the main difference from previous works [13, 15] is a few additional multilayer

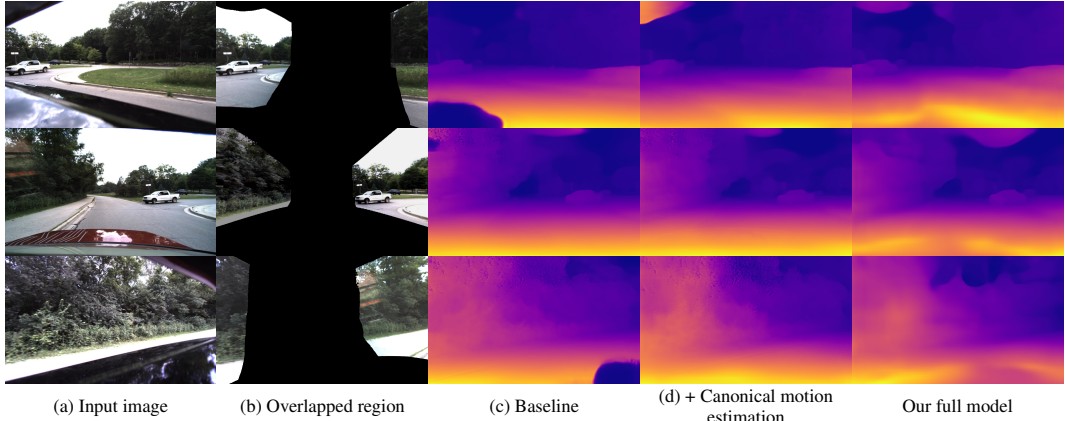

| (a) Input image | (b) Overlapped region | (c) Baseline | (d) + Canonical motion estimation | Our full model |

Figure 6: **Qualitative comparison from our ablation study**: *(a)* Given a set of images (*e. g.*, rear-left, rear, and rear-right), *(b)* estimated overlapped regions from nearby cameras, *(c)* depth results from our reproduced baseline, *(d)* depth results using the baseline model with our canonical motion estimation module, and our final results using both volumetric feature fusion and canonical motion estimation module.

Table 2: **Ablation study on our volumetric feature fusion and canonical fusion modules**: Our volumetric feature fusion scheme and canonical motion estimation module consistently improve the depth accuracy, substantially on *Abs Rel* and *Sq Rel* metrics.

| Volumetric feature fusion | Canonical motion | Abs Rel | Sq Rel | RMSE | RMSE log | $\delta < 1.25$ | $\delta < 1.25^2$ | $\delta < 1.25^3$ |
|---|---|---|---|---|---|---|---|---|
| *(Our reproduced baseline)* | | 0.228 | 4.409 | 13.433 | 0.342 | **0.687** | **0.870** | **0.932** |
| ✓ | | 0.222 | 4.055 | 13.474 | 0.348 | 0.682 | 0.862 | 0.928 |
| | ✓ | 0.222 | 3.969 | 13.492 | 0.344 | 0.677 | 0.865 | 0.931 |
| ✓ | ✓ | **0.218** | **3.660** | **13.327** | **0.339** | 0.674 | 0.862 | **0.932** |

Table 3: **Comparison result of different feature encoding methods**: To aggregate surround-view image features into a voxel space, we compare three different ways, averaging features (Avg voxel), applying 3D convolution layers (3D conv), or using MLPs (Proposed). Our proposed approach using MLPs demonstrates better depth accuracy than the other methods.

| Depth fusion | Abs Rel | Sq Rel | RMSE | RMSE log | $\delta < 1.25$ | $\delta < 1.25^2$ | $\delta < 1.25^3$ |
|---|---|---|---|---|---|---|---|
| Avg voxel | 0.224 | 4.150 | 13.464 | 0.346 | 0.672 | **0.865** | 0.931 |
| 3D conv | 0.228 | 4.179 | 13.474 | 0.348 | **0.680** | 0.862 | 0.924 |
| **Proposed (MLPs)** | **0.218** | **3.660** | **13.327** | **0.339** | 0.674 | 0.862 | **0.932** |

perceptrons with volumetric feature representation while keeping the similar conventional encoder and decoder networks. Furthermore, we show the benefit of the volumetric feature representation for the camera motion module; globally reasoning the camera motion benefits the depth estimation. Fig. 6 further provides qualitative results of each setting in the ablation study and demonstrates qualitative gains of each contribution.

**Volumetric feature encoder.** Table 3 shows the comparison result using different approaches to aggregate surround-view image features into a shared voxel space. We compare three different methods that are commonly used to process features in the voxel space. *Avg voxel* simply averages features that are accumulated at the same voxel coordinate. *3D conv* applies two sequential 3D convolutional blocks to the volumetric feature. Each 3D convolutional block is composed of a 3D convolutional layer, batch normalization, and LeakyReLU, in a consecutive order. Our method, using *MLPs*, clearly outperforms the rest two methods. Compared to 3D convolution, we find that *MLPs* more effectively learn to pool or weight multiple features on overlap regions.

In Fig. 7, we further visualize error maps of each method (*Avg voxel*, *3D conv*, *MLPs*) including the baseline, FSM [15]. The results provide clear evidence that the volumetric feature approach takes full advantage of overlap regions compared to the [15]. Furthermore, our MLP-based method shows the highest accuracy in overall regions including moving objects, compared to *Avg voxel* or *3D conv*.

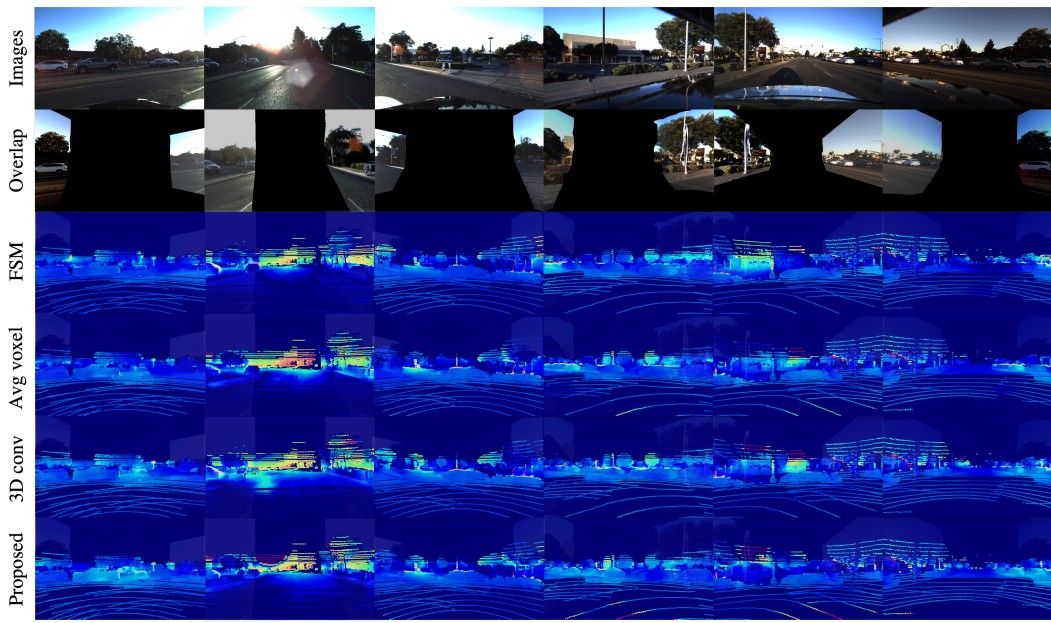

Figure 7: **Comparison on different volumetric feature encoding methods**: Qualitative comparison result using Abs Rel error map to visualize the model performance by regions. Methods using volumetric features (Avg voxel, 3D conv, Proposed) show accuracy improvement especially on overlap regions compared to FSM [15].

## 5 Conclusions

We proposed a novel volumetric feature representation for self-supervised surround-view depth estimation. The volumetric feature representation aggregates image features from surround-view images, encodes 3D information, and then is used to estimate a depth map at each view and canonical motion between two temporally consecutive images. Furthermore, by projecting the volumetric feature at arbitrary rotated views, our method can also synthesize a depth map at the novel view, additionally with varying focal lengths. Our method outperforms the state-of-the-art surround-view depth method. The ablation study successfully validates our contributions as well as design choices on the volumetric feature encoding. In future work, we expect our volumetric feature representation can benefit other computer vision tasks such as object detection, image segmentation, and motion estimation in the surround-view camera setup.

**Limitation.** Our volumetric feature representation is defined in a 3D cubic space; it needs to take the trade-off between memory usage and the resolution into consideration. Because we focus on embedding 3D information to a novel representation, sometimes some blurred regions occurs in the depth map (e. g., in Fig. 1 and Fig. 6). The usage of cylindrical or spherical coordinates would improve the efficiency of the memory usage. Furthermore, some artifacts sometimes appear in the synthesized depth map in Fig. 1 where no input image feature is given. In future work, we will consider using additional regularization terms to preserve details of depth results as well as reduce the artifact.

**Potential negative societal impacts.** Methods for surround-view depth estimation require a large number of images for training, which may include images with privacy issues. To prevent this, the dataset should be precisely curated and be used for the research purpose only.

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
