# Self-Supervised Surround-View Depth Estimation with Volumetric Feature Fusion

## – Supplementary Material –

**Jung-Hee Kim** [*]
42dot Inc.
junghee.kim@42dot.ai

**Junhwa Hur** [*,†]
Google Research
junhwahur@google.com

**Tien Phuoc Nguyen** [†]
Hyundai Motor Group Innovation Center
tien.nguyen@hmgics.com

**Seong-Gyun Jeong**
42dot Inc.
seonggyun.jeong@42dot.ai

In this supplementary material, we provide details on evaluation metrics, details on our network architecture, a trade-off between computational cost and depth accuracy, additional qualitative results, depth accuracy on overlap regions, point cloud results on the DDAD dataset and nuScenes dataset, and the license of existing assets we used for our paper.

## A  Evaluation metric

To evaluate the depth accuracy, we use the error metric proposed by Eigen et al. [8]. Given a predicted depth map $d$ and ground-truth map $d^*$ with $N$ pixels, it computes:

- Absolute relative error (Abs Rel): $\frac{1}{|N|} \sum_{i \in N} \frac{|d_i - d_i^*|}{d_i^*}$

- Square relative difference (Sq Rel): $\frac{1}{|N|} \sum_{i \in N} \frac{\|d_i - d_i^*\|^2}{d_i^*}$

- Root mean square error (RMSE): $\sqrt{\frac{1}{|N|} \sum_{i \in N} \|d_i - d_i^*\|^2}$

- Root mean squared logarithmic error (RMSE log): $\sqrt{\frac{1}{|N|} \sum_{i \in N} \|\log (d_i) - \log (d_i^*)\|^2}$

- Accuracy with threshold ($\delta$): % of $d_i$ s.t $\max \left( \frac{d_i}{d_i^*}, \frac{d_i^*}{d_i} \right) = \delta < thr$

## B  Details on the Network Architecture

We provide further details on our network architecture with Table 3. For more information about the implementation, please refer to our source code. Our model uses only 1D/2D convolutions and MLPs; we do not use 3D convolution which is computationally heavy and consume extensive memory.

**Image encoder.**  We used pre-trained ResNet-18 [16] for the image encoder. After extracting a feature pyramid, we upsample and concatenate the last three feature maps, and apply $1 \times 1$ convolutional layer to reduce the number of channels to 256.

**Volumetric encoder.**  To encode a volumetric feature from an image feature that is concatenated with a depth value, we use MLPs with two layers where the numbers of nodes are 256 and 128. In the end, each voxel contains a 128-dimensional feature vector.

---

[*]denotes equal contribution. [†] This work has been done at 42dot Inc.

36th Conference on Neural Information Processing Systems (NeurIPS 2022).

Table 3: **Network details.**

×2 ↑: upsampling an input to the two times bigger resolution

| Module | Layer output | Layer operation | Input | Output size |
|---|---|---|---|---|
| Image encoder | feat1 | ResNet conv2d | Input image | $64 \times H/2 \times W/2$ |
| | feat2 | ResNet block1 | feat1 | $64 \times H/4 \times W/4$ |
| | feat3 | ResNet block2 | feat2 | $128 \times H/8 \times W/8$ |
| | feat4 | ResNet block3 | feat3 | $256 \times H/16 \times W/16$ |
| | feat5 | ResNet block4 | feat4 | $512 \times H/32 \times W/32$ |
| | feat6 | Upsample & concat. | [feat3, feat4, feat5] | $896 \times H/8 \times W/8$ |
| | img_feat | conv2d | feat6 | $256 \times H/8 \times W/8$ |
| Volumetric encoder | backproj_feat1 | Back-projection | img_feat | $256 \times Z \times Y \times X$ |
| | backproj_feat2 | Concat. with depth | [backproj_feat1, depth] | $257 \times Z \times Y \times X$ |
| | volumetric_feat | MLP | backproj_feat2 | $128 \times Z \times Y \times X$ |
| Depth fusion | proj_img_feat1 | Sampling & concat. | volumetric_feat | $(50 * 128) \times H/8 \times W/8$ |
| | proj_img_feat2 | MLP | proj_img_feat1 | $128 \times H/8 \times W/8$ |
| | depth_upconv1 | conv2d, ELU, ×2 ↑, conv2d, ELU | proj_img_feat2 | $64 \times H/4 \times W/4$ |
| | depth_upconv2 | conv2d, ELU, ×2 ↑, conv2d, ELU | depth_upconv1 | $32 \times H/2 \times W/2$ |
| | depth_upconv3 | conv2d, ELU, ×2 ↑, conv2d, ELU | depth_upconv2 | $16 \times H \times W$ |
| | depth output | conv2d, sigmoid | depth_upconv3 | $1 \times H \times W$ |
| Canonical motion estimation | collapsed_voxel | Voxel collapse | volumeteric_feat | $(128 * Z) \times Y \times X$ |
| | fused_pose_feat | Channel reduction | collapsed_voxel | $256 \times Y \times X$ |
| | squeeze_feat | conv2d, ReLU | fused_pose_feat | $256 \times Y \times X$ |
| | pose_feat1 | conv2d, ReLU | squeeze_feat | $256 \times Y/2 \times X/2$ |
| | pose_feat2 | conv2d, ReLU | pose_feat1 | $256 \times Y/4 \times X/4$ |
| | pose output | conv2d, avgpool | pose_feat2 | $6 \times 1 \times 1$ |

Table 4: **Effect of the voxel grid size on computational cost and depth accuracy**: A choice of voxel grid size causes a trade-off between computational cost and depth accuracy.

| Voxel grid size $[x, y, z]$ | Computational cost | | | Accuracy | | |
|---|---|---|---|---|---|---|
| | Parameters | FLOPs | GPU memory | Abs Rel | Sq Rel | $\delta < 1.25$ |
| $[150, 150, 15]$ | 15.5 M | 203 G | 6.04 G | 0.212 | 3.899 | 0.706 |
| $[130, 130, 13]$ | 15.5 M | 189 G | 4.74 G | 0.215 | 4.079 | 0.706 |
| $[80, 80, 8]$ | 15.5 M | 169 G | 4.41 G | 0.223 | 4.273 | 0.701 |

**Depth fusion module.** We uniformly sample 50 voxel features for each pixel ray, concatenate them, and use a MLP with two layers whose numbers of nodes are both 128. Then we apply the following operations for three times in the depth decoder: $3 \times 3$ conv, ELU, upscale (×2 resolution), and $3 \times 3$ conv. Then finally a $3 \times 3$ convolutional layer with a sigmoid activation is applied to the output inverse depth map.

**Canonical motion estimation module.** To construct features for the 2D convolutional layer, we reduce the $\mathbf{Z}$ dimension of the volumetric feature by voxel collapse operation, which concatenates $\mathbf{Z}$ dimension to $\mathbf{C}$ dimension. We then apply $1 \times 1$ convolutional layer with batch normalization and ReLU activation to reduce the channel to be $\mathbf{C}'$, where $\mathbf{C}$ and $\mathbf{C}'$ indicates 128 and 256, respectively. The features are then fed into the pose decoder that has the same structure as previous work [13].

## C   Effect of the voxel grid size on computational cost and depth accuracy

Table 4 shows how the voxel resolution affects the computational cost and depth accuracy, which causes a trade-off between them. Given a fixed size of voxel space that covers $100(m) \times 100(m) \times 30(m)$ range, we vary the voxel unit size (i. e., the length of one side of a voxel cubic) and the number of voxels along each axis accordingly. For this study, we use the depth-fusion-only model. The depth accuracy increases with a finer voxel resolution (i. e., smaller unit size with more voxels), along with the increasing GPU memory consumption. We believe the usage of spherical or cylindrical coordinate representation will improve the memory efficiency.

Despite more memory consumption than standard monocular depth approaches using 2D representation, we expect our volumetric representation can also benefit other image-based 3D perception tasks, such as 3D object detection, 3D semantic segmentation, 6D object pose estimation, or bird's-eye-view (BEV) segmentation, in the surround-view camera setup.

Table 5: **Accuracy analysis on overlap region**: We compare depth accuracy on overlap regions from nearby cameras.

| Depth fusion | Abs Rel | Sq Rel | RMSE | RMSE log | $\delta < 1.25$ | $\delta < 1.25^2$ | $\delta < 1.25^3$ |
|---|---|---|---|---|---|---|---|
| *(Our reproduced baseline)* | 0.239 | 4.299 | 13.300 | 0.352 | 0.658 | 0.856 | 0.926 |
| **Ours** | **0.237** | **4.219** | **12.782** | **0.346** | **0.665** | **0.858** | **0.929** |

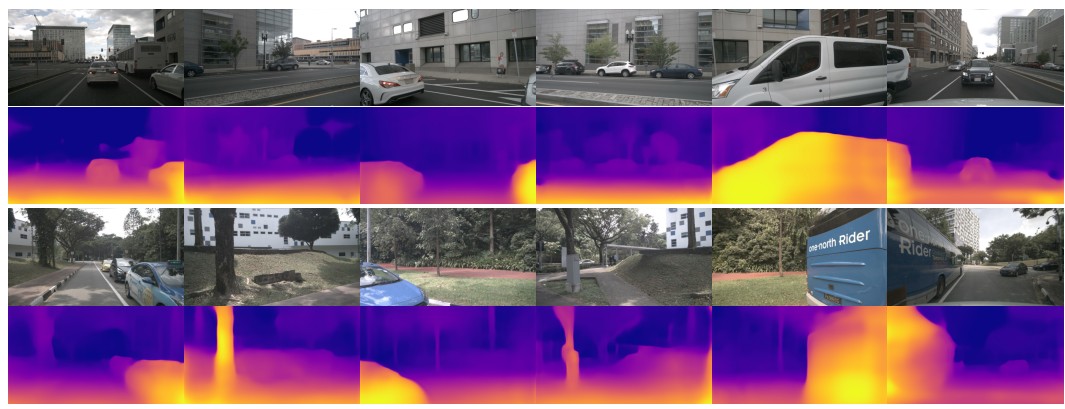

Figure 7: **Our results on nuScenes dataset**: Our method accurately estimates depth under various conditions.

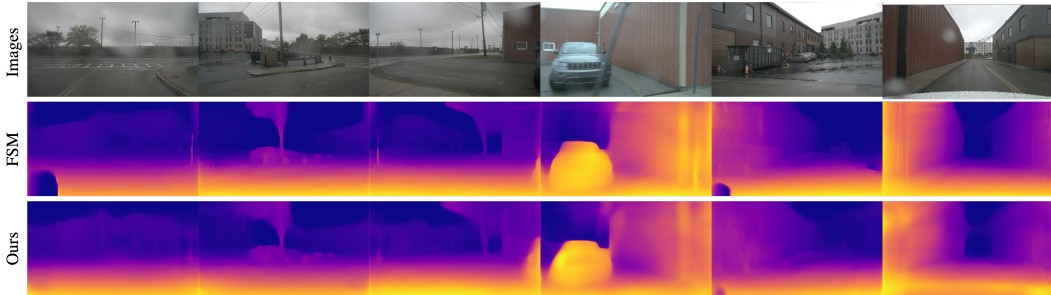

Figure 8: **Comparison results on nuScenes dataset**: Comparison results with nuScenes dataset using the baseline model and our model.

# D    Additional Experimental Results

### D.1    Qualitative results on the nuScenes dataset

Fig. 8 visualizes comparison between FSM [15] and our proposed model. The depth results give evidence that our volumetric feature encoding enhances consistency and connectivity between different views. Our model gives results that can smoothly transit when FSM model creates holes in overlap regions.

### D.2    Improvement on overlap regions

Our model takes advantage of overlapping regions as the same features are shared with neighboring cameras by using our volumetric features. Therefore, we have conducted additional experiments to analyze depth estimation accuracy in overlapping regions. The overlapping regions were extracted as shown in the Fig. 6 on the main paper. As can be seen with the Table 5 and Fig. 8, our model demonstrates improved accuracy and consistent depth in the overlapping regions.

### D.3    Point cloud visualization using surrounding view cameras

We further demonstrate our contributions using the point cloud reconstruction results. Point cloud results are evaluated on both DDAD and nuScenes dataset as shown in Fig. 9 and Fig. 10 respectively.

As we focus on learning to merge features on overlap regions, we've obtained results that show enhanced alignment between different viewpoints in overlap regions, which supports our contribution towards utilizing overlap regions and global reasoning.

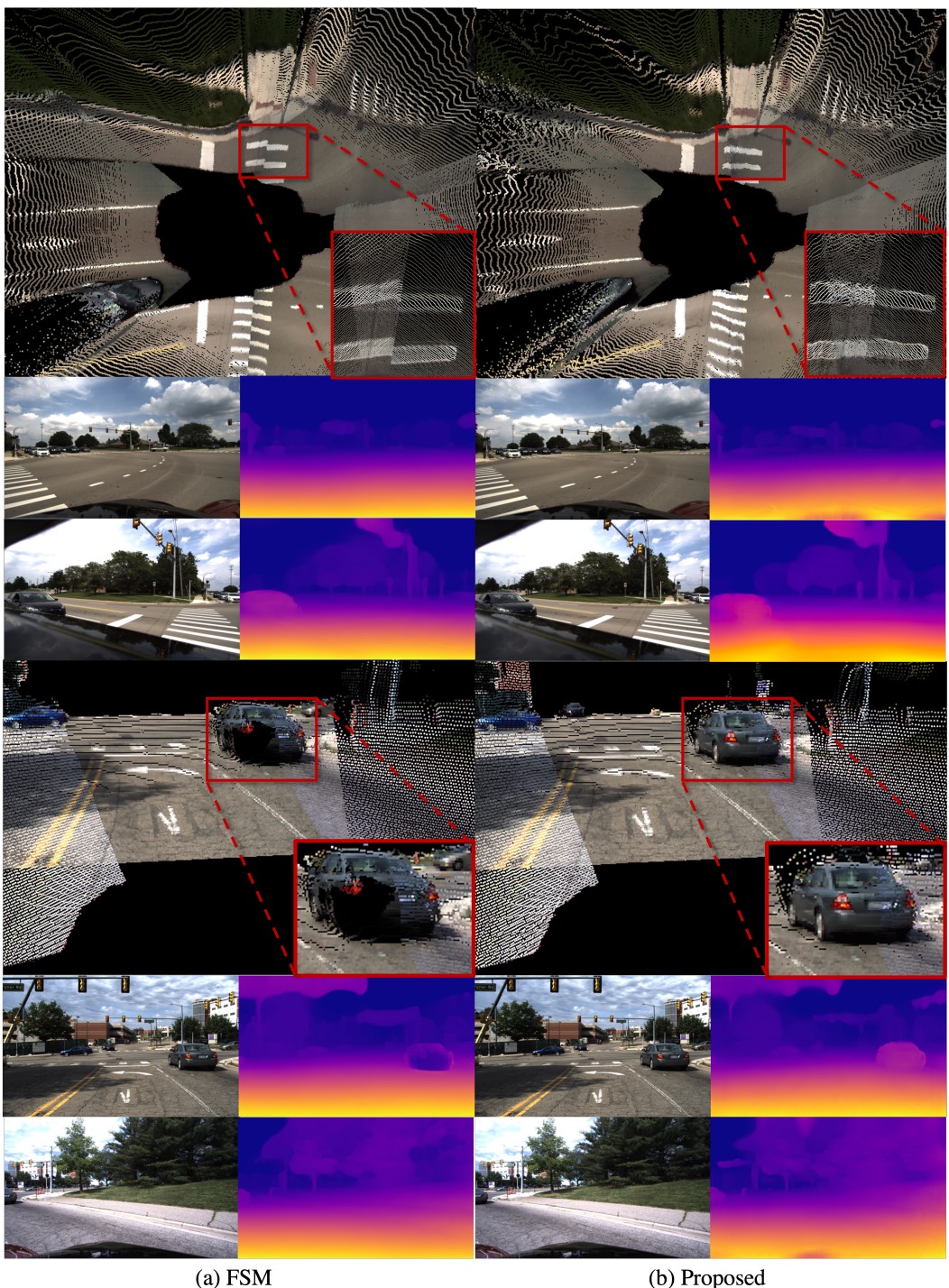

(a) FSM        (b) Proposed

Figure 9: **Point cloud reconstruction comparison results on the DDAD dataset**

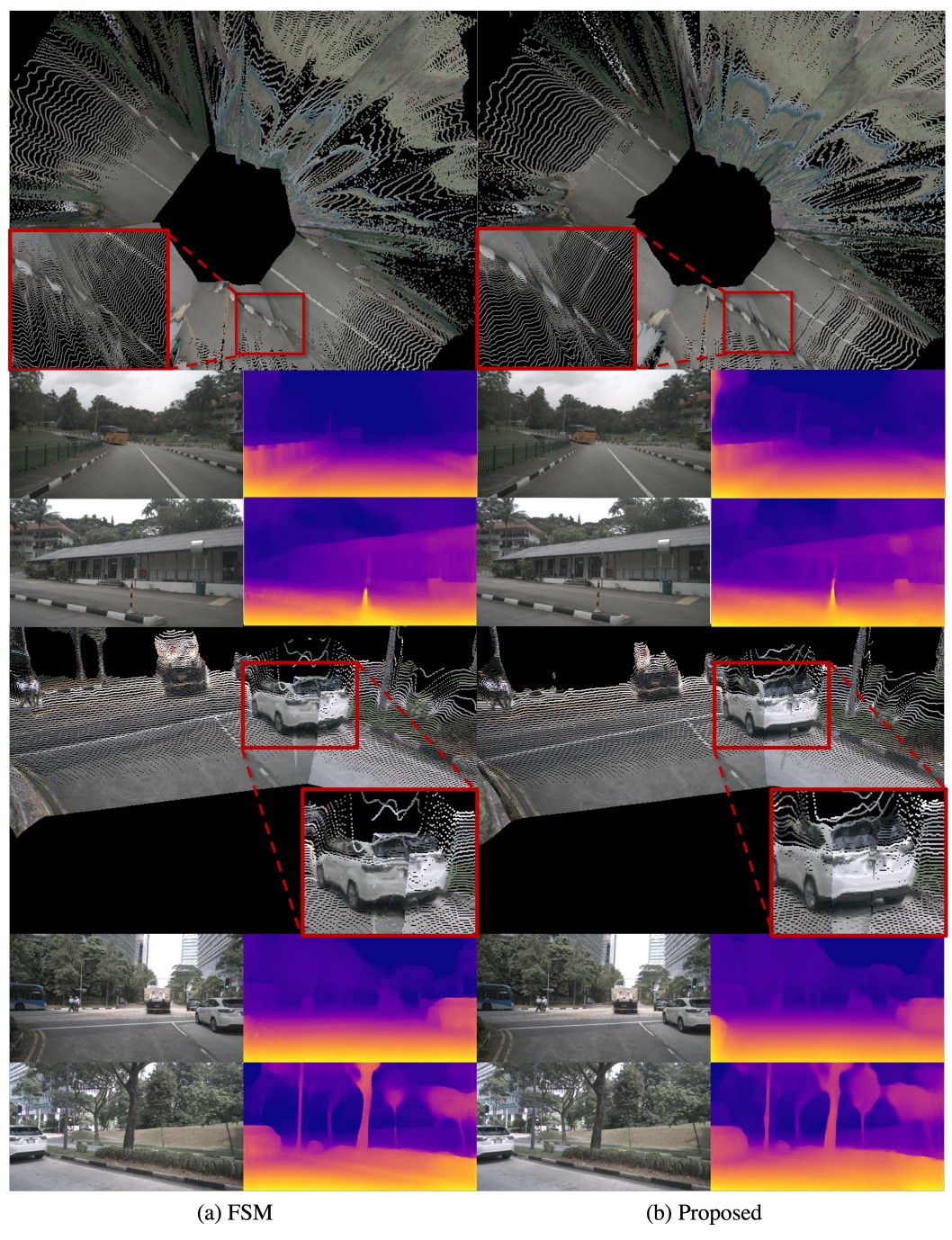

(a) FSM                                    (b) Proposed

Figure 10: **Point cloud reconstruction comparison results on the nuScenes dataset**

# E  License of the Existing Assets

For the experiment, we use the DDAD[2] [14] and nuScenes[3] [2] dataset. Both datasets are under a Creative Commons Attribution-NonCommercial-ShareAlike 4.0 International License (CC BY-NC-SA 4.0). Our implementation is partially based on PackNet-SfM[4] [14] (MIT license) and DGP[5] (MIT license). PyTorch [31] is under BSD License.

---

[2] `https://github.com/TRI-ML/DDAD`
[3] `https://www.nuscenes.org/terms-of-use`
[4] `https://github.com/TRI-ML/packnet-sfm`
[5] `https://github.com/TRI-ML/dgp`