# OpenReview forum: "Self-supervised surround-view depth estimation with volumetric feature fusion"
_NeurIPS.cc/2022/Conference — NeurIPS 2022 Accept_

### Official Review · Reviewer_ZBww · 2022-07-03

**Rating:** 5
**Confidence:** 4
**Soundness:** 3 good
**Presentation:** 3 good
**Contribution:** 3 good

**Summary:**

This paper introduces a self-supervised depth estimation method based on a unified volumetric feature representation encoded from surround-view. The proposed method consists of three parts. First, the surround-view feature fusion module generates a unified volumetric feature from the extracted multi-view 2D image features. Second, the depth fusion module reconstructs the depth map given an input camera viewpoint. Last, the global motion of the canonical camera is estimated by assuming static camera extrinsic. Experiments on DDAD and nuSecens datasets achieved improved results over existing methods.


**Questions:**

- More details for the construction of the volumetric feature.
- Effect of volume resolution on memory consumption and depth estimation.

**Limitations:**

The authors have discussed the limitation and potential negative societal impact of their work.

**Strengths And Weaknesses:**

Strengths:
- The idea of using a volumetric feature to encode information from surround-view images seems interesting and logical.
- With the construction of volumetric feature and depth fusion, the proposed method achieves improved results on DDAD and nuScenes datasets.
- Ablation study has been provided to verify the effectiveness of depth fusion and canonical motion estimation.


Weaknesses:
- The core of this method is the construction of the unified volumetric feature to estimate the surround-view depth and canonical camera motion. However, some details for constructing a unified volumetric feature are missing. This method simply copies the image feature to all voxels went through by the corresponding pixel ray. In this case, some voxels may have been passed by multiple pixel rays. It is unclear how to handle multi-feature collisions.
- Related to the first question, why different MLPs are used to fuse the overlap features and non-overlap features in the volumetric feature (see Line 132)? No experiments are shown to verify this strategy.
- Since this method encodes the feature in a 3D volume, the effect of volume resolution on the memory consumption and depth estimation should be discussed.

Overall I think the proposed idea is interesting, but I have some questions for the volumetric feature construction part. I would give a borderline accept rating at this stage.

---

> ### Author Response · Authors · 2022-08-02
> **Response to Reviewer ZBww**
>
> Thank you for your positive feedback and suggestions!
>
> ***
>
> **● Details for constructing a unified volumetric feature. How to handle multi-feature collisions?**
>
> We provide clearer details here and revised the main paper accordingly (`Sec. 3.2` and `Fig. 3`).
>
> For each voxel, we find a corresponding pixel coordinate $\mathbf{p}(w, h)$ by projecting the voxel into a pixel coordinate.
> Then, we bilinearly interpolate an image feature $\mathbf{F}(\mathbf{p})$ at the projected pixel $\mathbf{p}(w, h)$ to handle the sub-pixel location, and allocate the sampled feature to the voxel.
> Because the sampled image feature $\mathbf{F}(\mathbf{p})$ contains high-level information along its pixel ray, we extract the local 3D feature at the voxel $(x, y, z)$ by concatenating the feature with a depth value of the voxel (i.e., depth positional encoding) and pass it through an MLP.
> In this way, voxels that gets through by a single pixel ray have their individual 3D feature extracted by the MLP instead of having the same image feature.
>
> Yes, some voxels in overlap regions do correspond to multiple image features.
> To merge the multiple features, we use the other MLP, for the overlap regions, that learn to merge the features.
> We concatenate the multiple image features and pass them through the MLP to obtain the merged feature (ref, `Fig. 3`).
>
> ***
>
> **● Why different MLPs are used to fuse the overlap features and non-overlap features in the volumetric feature?**
>
> As explained above, the main reason is to effectively merge the multiple features for the voxels that correspond to multiple image features.
> To validate the benefit of our technical design, we provide a comparison using different approaches to encode volumetric features:
>
> | Method  | Abs Rel | Sq Rel | RMSE | Log RMSE | $\delta < 1.25$ | $\delta < 1.25^2$ |$\delta < 1.25^3$|
> |-----------------------|:---------:|:--------:|:----:|:--------:|:----------:|:------------:|:------------:|
> | **MLPs (Ours)**       | **0.218** | **3.660** | **13.327** |  **0.339**   | 0.674      | 0.862        |   **0.932**    |
> | Avg voxel             |  0.224    |   4.150  |13.464|    0.346 | 0.672 |        **0.865** |   0.931    |
> | 3D conv               | 0.228     |   4.179  |13.474|  0.348   | **0.680**      |  0.862       |   0.924    |
>
> We compare three different methods that are commonly used to handle features in voxel space.
> *Avg voxel* simply averages features that are accumulated at the same voxel coordinate.
> *3D conv* applies two sequential 3D convolutional blocks (with BatchNorm and LeakyReLU) to the accumulated 3D volumetric features.
> The experiment is conducted on the DDAD dataset, by evaluating metric-scale depth.
>
> Our method, using *MLPs*, clearly outperforms the rest two methods.
> Compared to 3D convolution, we find *MLPs* more effectively learn to pool or weight multiple features on overlap region.
>
> We include this discussion in `Sec. C` in supplementary with additional qualitative results.
>
> ***
>
> **● Effect of voxel resolution on computational cost and depth accuracy**
>
> We kindly ask to refer to the common response **([Link](https://openreview.net/forum?id=0PfIQs-ttQQ&noteId=aQJZz7GDd1m))** on the trade-off between computational cost and depth accuracy by using different voxel resolution.
>
> ***
>
> We hope our responses address your questions. Please let us know if you have further questions or concerns! We are happy to answer them.

---

### Official Review · Reviewer_9Ef1 · 2022-07-11

**Rating:** 5
**Confidence:** 4
**Soundness:** 3 good
**Presentation:** 3 good
**Contribution:** 2 fair

**Summary:**

This paper proposes a novel surround-view depth estimation, and project features of different views into a volumetric feature spaces, where overlapped views can enhance each other and get better depth maps. The pipeline also enables novel-view depth synthesis. The problem setting may be useful for autonomous driving.

The experiments are insufficient.

**Questions:**

- Is it possible for the proposed method to compare with monocular methods? There are many common benchmarks for monocular depths.
- Would the cubic space introduce some artifacts and distortions comparing to spherical spaces?
- As a self-supervised method, I assume there is no GT depth needed? Is that so? Is it possible to train on real data without GT?

**Limitations:**

Yes.

**Strengths And Weaknesses:**

+ Paper is easy to follow.
+ The problem setting is novel.

- The depth map quality is on par with monocular methods.
- No visualizations of different methods on DDAD and nuScenes are provided. From Figure 5, the depth maps are similar to existing monocular methods in quality.
- In Table 2, depth fusion is not evaluated individually.

---

> ### Author Response · Authors · 2022-08-02
> **Response to Reviewer 9Ef1**
>
> Thank you for your positive and valuable feedback!
>
> ***
>
> **● Additional qualitative results**
>
> We kindly ask to refer to the common response **([Link](https://openreview.net/forum?id=0PfIQs-ttQQ&noteId=aQJZz7GDd1m))** on the additional qualitative results.
> The revised supplementary material includes additional results.
>
> ***
>
> **● Ablation study in Table 2**
>
> We provide the complete ablation study including the depth-fusion-only model:
>
> |      Canonical motion     | Depth fusion | | Abs Rel | Sq Rel |  RMSE  | RMSE log | $\delta < 1.25$ | $\delta < 1.25^2$ | $\delta < 1.25^3$ |
> |:-------------------------:|:-------------:|-|:-------:|:------:|:------:|:--------:|:----------:|:------------:|:------------:|
> | (Baseline) |          |   |  0.228  |  4.409 | 13.433 |   0.342  |    0.687   |     0.870    |     0.932    |
> |             ✔             |             | |  0.222  |  3.969 | 13.492 |   0.344  |    0.677   |     0.865    |     0.931    |
> |                           |       ✔     | |  0.222  |  4.055 | 13.474 |   0.348  |    0.682   |     0.862    |     0.928    |
> |             ✔             |       ✔     | |  0.218  |  3.660 | 13.327 |   0.339  |    0.674   |     0.862    |     0.932    |
>
> The depth-fusion-only model (3rd row) improves the accuracy of the baseline by 2.64% in the *Abs Rel* metric and by 8.03% in the *Sq Rel* metric.
> The depth fusion module also demonstrates consistent accuracy improvement on the canonical-motion-only model (i.e., from the 2nd row to the 4th row).
> In the main paper, we updated `Table 2` and the text accordingly.
>
> ***
>
> **● Is it possible for the proposed method to compare with monocular methods?**
>
> Yes, it's possible to compare with monocular methods, and our method would perform on-par with MonoDepth2 [13] or FSM [15] as shown in Table 1 in FSM [15].
> However, our method mainly focuses on a surround-view camera setup and introduces a volumetric representation that effectively fuses feature maps from each view into a unified feature space.
> Our proposed representation can be adapted to any monocular depth methods that follow standard encoder-decoder architectures.
>
> ***
>
> **● Would the cubic space introduce some artifacts and distortions comparing to spherical spaces?**
>
> We don't observe any cubic-shaped artifacts or distortions, but see some artifacts in synthesized depth maps due to empty voxels where no input image feature is given.
> We expect that using stronger regularization or learned initialization values can resolve the problem, which warrants further research.
> As future work, we expect that using a spherical-coordinate-based voxel space can improve memory efficiency.
>
> ***
>
> **● As a self-supervised method, I assume there is no GT depth needed? Is that so? Is it possible to train on real data without GT?**
>
> Yes, our approach is a purely self-supervised method without requiring ground-truth depth.
> We demonstrate self-supervised experiments on the real-image datasets, DDAD and nuScenes.
> Our method only assumes known camera intrinsics and extrinsics with a small overlap between spatially neighboring images.
> Our method is flexible with the number of input views.
>
> ***
>
> We hope our responses address your questions. Please let us know if you have further questions or concerns! We are happy to answer them.

---

### Official Review · Reviewer_K314 · 2022-07-13

**Rating:** 6
**Confidence:** 3
**Soundness:** 2 fair
**Presentation:** 2 fair
**Contribution:** 2 fair

**Summary:**

The authors propose a self-supervised approach for the task of surround-view depth estimation. 2D image features are un-projected onto a shared 3D volume, which is later queried for decoding depth maps at target views. Similarly, pose changes of the canonical frame is also decoded from volumetric feature pooled from all 2D views. Evaluations conducted on DDAD and nuScenes show competitve results compared to FSM and others.

**Questions:**

It's important to address the closely related prior work mentioned above and highlight any new contributions.

Also, I feel the paper needs to have a more convincing analysis of the differences compared to FSM and others, e.g. examples highlighting typical qualitatively improvements.

**Limitations:**

Adequately addressed.

**Strengths And Weaknesses:**

__Strengths__

Because of the use of a volumetric feature representation, features from images are aggregated in a shared space. This should help produce more consistent depth maps across views.

Also because of the use of a volumetric feature representation, the proposed model has the unique capability of predicting depth maps for views not among the inputs.


__Weaknesses__

A few relevant works are not included or discussed sufficiently. The way 2D features are un-projected onto a shared 3D volume aggregation has been used in many places, e.g. [1], [2], [3].

[1] Atlas: End-to-End 3D Scene Reconstruction from Posed Images, ECCV 2020.
[2] Lift, Splat, Shoot: Encoding Images from Arbitrary Camera Rigs by Implicitly Unprojecting to 3D, ECCV 2020.
[3] Learning a Multi-View Stereo Machine, NIPS 2017.

In particular, [3] is very similar as it also fuses 2D features from multiple views on a shared 3D volume which is later projected to 2D views for depth estimation.

The quality advantage over existing methods is not convincing:
* Although slightly better than FSM quantitatively in some case, the gap is usualy quite minimal and sometimes non-existence.
* Qualitative comparisons (e.g. in Fig.6) are not showing any obvious improvements.
* How do the depth maps look if viewed as point cloud? This can also provide insights if the proposed method has a better agreement between adjacent views.

Because of the volumetric feature representation, there is a trade-off between resolution and compute. This is not a concern for image-space models.

A few more minor issues:
* The "outer product" operation in Fig.3 is not mentioned anywhere in the text and contradicts the architecture given in the supplementary.
* The loss terms are not given precise definitions.
* L.257 ("provide our attempts to reproduce and discussions in the supplementary material.") -- not found in supplementary.

---

> ### Author Response · Authors · 2022-08-02
> **Response to Reviewer K314**
>
> Thank you for your constructive and valuable feedback!
>
> ***
>
> **● Additional qualitative results**
>
> We kindly ask to refer to the common response **([Link](https://openreview.net/forum?id=0PfIQs-ttQQ&noteId=aQJZz7GDd1m))** on the additional qualitative results.
> The revised supplementary material includes additional results.
>
> ***
>
> **● Trade-off between computational cost and depth accuracy by voxel resolution**
>
> We kindly ask to refer to the common response **([Link](https://openreview.net/forum?id=0PfIQs-ttQQ&noteId=aQJZz7GDd1m))** on the trade-off between computational cost and depth accuracy by using different voxel resolutions.
>
> ***
>
> **● Minor issues - clarification**
>
> - **Outer product in Fig. 3**: Thank you! It's our mistake; the architecture description in the supplementary is indeed correct.
> We updated `Fig. 3` more precisely with clearer technical details in the paragraph *Volumetric feature encoding* in `Sec. 3.2` main paper.
>
> - **Loss term**: Thank you for your feedback! We updated precise definitions on each loss term in `Sec. 3.5` in the main paper.
>
> - **Baseline reproduction**: Thank you for the comment! We included the following clarification in `Sec. 4.1` in the main paper.
>   - For the baseline (FSM [15]) reproduction, we found two additional components were crucial: the intensity distribution alignment (`Eq. (6)`) and focal length normalization from Cam-Conv [CAM].
>   -  The focal length normalization corrects the scale of output depth by input camera's focal length so that the network learns to output a consistent depth scale regardless of different camera focal lengths. Without the normalization, we found that FSM [15] is not stably trained on the DDAD dataset [14], where two types of cameras with different focal lengths are used.
>
>
>
> ***
>
> **● It's important to address closely related prior work and highlight new contributions**
>
> The main contribution is to propose a volumetric feature approach suitable for surround-view depth estimation.
>
> The basic idea of back-projecting multi-view features into a shared space has been presented in other tasks such as multi-view stereo [LSM, MVSNet, UniMVS, ATLAS, RayNet] or 3D semantic segmentation [ATLAS, 3DMV].
> However, those methods mainly target a matching task for object-level or static indoor scene reconstruction using computationally-heavy 3D convolution, which is not suitable for our task.
>
> For the surround-view depth estimation in dynamic outdoor scenes, we design to use light-weighted MLP layers that fuse features in the small overlap between images and extract local 3D features from image ray features into each voxel coordinate.
>
> Our ablation study in `Table 4 and Sec. C (supplementary)` shows the accuracy benefit of the MLP layers over 3D convolution [LSM, MVSNet, UniMVS] or average pooling.
> Our representation also benefits canonical camera motion estimation in the surround-view setup.
> Unlike Kar et al. [LSM], our method demonstrates synthesizing depth maps at the novel view, facilitated by our depth synthesis loss in `Eq. (5a)`.
>
> In the surround-view setup, some approaches [LSS, FIERY] demonstrate aggregating 3D features into a Bird's-eye-view (BEV) space for semantic segmentation or object detection.
> However, the BEV-based representation does not precisely preserve 3D information.
> In contrast, our approach presents a voxel-based representation that is more suitable for not only depth and canonical pose estimation but also  seamless depth map synthesis at novel views.
> We expect that our representation can enrich other 3D tasks for the dynamic surround-view setup.
>
> ***
>
> We hope our responses address your questions. Please let us know if you have further questions or concerns! We are happy to answer them.
>
> ***
>
> **Reference**
>
> - [3DMV] Dai and Nießner, "3DMV: Joint 3D-Multi-View Prediction for 3D Semantic Scene Segmentation", ECCV 2018
> - [ATLAS] Murez et al., "Atlas: End-to-End 3D Scene Reconstruction from Posed Images", ECCV 2020
> - [CAM] Facil et al., "CAM-Convs: Camera-Aware Multi-Scale Convolutions for Single-View Depth", CVPR 2019
> - [DV] Sitzmann et al., "DeepVoxels: Learning Persistent 3D Feature Embeddings", CVPR 2018
> - [FIERY] Hu et al., "FIERY: Future Instance Prediction in Bird's-Eye View from Surround Monocular Cameras", ICCV 2021
> - [LSM] Kar et al., "Learning a Multi-View Stereo Machine", NIPS 2017
> - [LSS] Philion and Fidler, "Lift, Splat, Shoot: Encoding Images from Arbitrary Camera Rigs by Implicitly Unprojecting to 3D", ECCV 2020
> - [MVSNet] Yao et al., "MVSNet: Depth Inference for Unstructured Multi-view Stereo", ECCV 2018
> - [RayNet] Paschalidou et al., "Learning Volumetric 3D Reconstruction with Ray Potentials", CVPR 2018
> - [UniMVS] Peng et al., "Rethinking Depth Estimation for Multi-View Stereo: A Unified Representation", CVPR 2022

---

### Author Response · Authors · 2022-08-02
**Response to all reviewers**

We sincerely thank all the reviewers for their time and constructive comments, which help improve our work.
Here, we address the common questions on additional qualitative results and the trade-off between memory consumption and depth accuracy.
We leave an individual response to each review below.
We also incorporated all our responses in the main paper and the supplementary material by including additional qualitative results, further clarification on the technical details, additional ablation studies, and discussions.

***

**● [K314 and 9Ef1] Additional qualitative results**

In order to highlight our contributions over the baseline, we provide additional qualitative results and point cloud visualization in the supplementary.

Our model demonstrates better accuracy especially on overlap regions, due to the usage of the volumetric feature representation shared by all views.
As in `Table 6` in supplementary, our method outperforms the baseline (FSM [15]) in the overlap regions on all metrics.

In `Fig. 10` and `Fig. 11` in supplementary `Sec. E.3`, the point cloud visualization clearly shows the evident advantage of our method in overlap region over FSM [15].
Our method demonstrates much better agreement between adjacent views.

Besides as shown in `Table 1` in the main paper, our method outputs more accurate metric-scale depth than FSM [15] on both DDAD and nuScenes datasets, substantially outperforming FSM [15] on *Abs Rel* and *Sq Rel* metrics.

***

**● [K314 and ZBww] Trade-off between computational cost and depth accuracy by voxel resolution**

We demonstrate how the voxel resolution affects computational cost and depth accuracy, which causes a trade-off between them:

| Voxel resolution $[x, y, z]$ | Abs Rel | Sq Rel | $\delta < 1.25$ |   |  #Param.   |  FLOPs   | GPU Memory |
|------------------------|:---------:|:--------:|:----------:|:---------:|--|:--------:|:----------:|
| [150, 150, 15]         |   0.212   |   3.899  |    0.706   |  |   15.5M    |  203G    |   6.04G    |
| [130, 130, 13]         |   0.215   |   4.079  |    0.706   |  |   15.5M    |  189G    |   4.74G    |
| [80, 80, 8]            |   0.223   |   4.273  |    0.701   |  |   15.5M    |  169G    |   4.41G    |

Given a fixed size of voxel space that covers $400 (m) \times 400 (m) \times 40 (m)$ range, we vary the voxel unit size (i.e., the length of one side of a voxel cubic) and the number of voxels along each axis accordingly.
For this study, we trained the depth-fusion-only model on the DDAD dataset and evaluated it on the test split.
The depth accuracy increases with a bigger voxel resolution (i.e., smaller unit size with more voxels), along with the increasing GPU memory consumption.
Our final model uses a smaller voxel unit size while reducing the voxel coverage range, in order to reduce computational cost without degrading the accuracy too much.

We believe the usage of spherical or cylindrical coordinate representation will improve the memory efficiency by having a varying voxel unit size according to the distance to the camera, which we will investigate in future work.

Despite more memory consumption than standard monocular depth approaches using 2D representation, we expect our volumetric representation can also benefit other image-based 3D perception tasks, such as 3D detection, 3D segmentation, or 6D pose estimation in the surround-view camera setup.
We prospect our model can also serve as a pretrained model for those 3D vision tasks.

This discussion is included in `Sec. D` in the revised supplementary material.

---

### Meta-Review · Area_Chair_3tWB · 2022-08-23

**Recommendation:** Accept
**Confidence:** Certain

**Metareview:**

Initially, the paper had mixed reviews (455).  The major concerns from the reviews were:

1. missing refs about unprojection. (K314)
2. quality advantage is not convincing, slightly better than FSM, while qualitative results show not obvious improvements. (K314)
3. visualize the depth maps as point clouds (K314)
4. what is the trade-off between resolution/memory, computation, and depth estimation? (K314, ZBww)
5. insufficient experiments (9Ef1)
6. comparison with monocular methods (9Ef1)
7. cubic vs spherical space? (9Ef1)
8. can it be trained on real data w/o GT supervision? (9Ef1)
9. how to handle collision of multiple pixel rays? (ZBww)
10. why use different MLPs to fuse "overlap" and "non-overlap" features? No ablation study on this. (ZBww)

The authors wrote a response to address these concerns, providing more qualitative results and ablation studies, as well as further explanations.  The reviewers were satisfied with the response, and K314 upgraded their rating to 6, while other reviewers maintained 5s. The reviewers appreciated the novel problem and the solution that can produce more consistent depth maps across views, and also synthesize depth maps in novel views. After reading the paper, the AC agrees with the reviewers, noting that the paper  addresses the limitations of the problem setup of previous work [13], thus developing a new line of research.  Thus, the AC recommends accept. The authors should prepare a revised version of the paper according to the reviews, rebuttal, and discussion.

**Award:**

No

---

### Decision · Program_Chairs · 2022-09-14

Accept